# Defining and quantifying fatigue in the rugby codes

Mitchell Naughton[1,2,3], Tannath Scott[4,5], Dan Weaving[4]*, Colin Solomon[1,2], Scott McLean[2]

1 School of Health and Behavioural Sciences, University of the Sunshine Coast, Sippy Downs, Queensland, Australia, 2 Centre for Human Factors and Sociotechnical Systems, University of the Sunshine Coast, Sippy Downs, Queensland, Australia, 3 Applied Sports Science and Exercise Testing Laboratory, University of Newcastle, Ourimbah, New South Wales, Australia, 4 Carnegie Applied Rugby Research Centre, Leeds Beckett University, Leeds, West Yorkshire, United Kingdom, 5 School of Health Sciences and Social Work, Griffith University, Gold Coast, Queensland, Australia

* d.a.weaving@leedsbeckett.ac.uk

## Abstract

The rugby codes (i.e., rugby union, rugby league, rugby sevens [termed 'rugby']) are team-sports that impose multiple complex physical, perceptual, and technical demands on players which leads to substantial player fatigue post-match. In the post-match period, fatigue manifests through multiple domains and negatively influences recovery. There is, however, currently no definition of fatigue contextualised to the unique characteristics of rugby (e.g., locomotor and collision loads). Similarly, the methods and metrics which practitioners consider when quantifying the components of post-match fatigue and subsequent recovery are not known. The aims of this study were to develop a definition of fatigue in rugby, to determine agreement with this common definition of fatigue, and to outline which methods and metrics are considered important and feasible to implement to quantify post-match fatigue. Subject matter experts (SME) undertook a two-round online Delphi questionnaire (round one; n = 42, round two; n = 23). SME responses in round one were analysed to derive a definition of fatigue, which after discussion and agreement by the investigators, obtained 96% agreement in round two. The SME agreed that fatigue in rugby refers to a reduction in performance-related task ability which is underpinned by time-dependent negative changes within and between cognitive, neuromuscular, perceptual, physiological, emotional, and technical/tactical domains. Further, there were 33 items in the neuromuscular performance, cardio-autonomic, or self-report domains achieved consensus for importance and/or feasibility to implement. Highly rated methods and metrics included countermovement jump force/power (neuromuscular performance), heart rate variability (cardio-autonomic measures), and soreness, mood, stress, and sleep quality (self-reported assessments). A monitoring system including highly-rated fatigue monitoring objective and subjective methods and metrics in rugby is presented. Practical recommendations of objective and subjective measures, and broader considerations for testing and analysing the resulting data in relation to monitoring fatigue are provided.

**Data Availability Statement:** All data is included either in the figures and tables within the paper, or in the Supporting information documents attached.

**Funding:** The author(s) received no specific funding for this work.

**Competing interests:** The authors have declared that no competing interests exist.

## Introduction

The rugby codes (i.e., rugby union, rugby league, rugby sevens [hereafter termed 'rugby']) are team-sports that impose multiple complex physical, perceptual, and technical demands on players. In rugby, there has been a proliferation of methods and metrics to quantify and monitor each specific component of training and match-play through technologies such as global positioning systems (GPS), accelerometers, and heart-rate telemetry to quantify the output of the player on the field. For example, the frequency, intensity, and duration of locomotor and collision actions (termed the external load), and the players psycho-physiological responses (termed the internal load) to the completed external loads [1–3].

In addition to improved quantification of match and training external and internal loads, there has been an increased recognition of the fatiguing effects of elevated match-related external and internal loads, and the negative impact of the associated fatigue may have on recovery into the subsequent training week [4, 5]. Aben et al. [4] identified that match-play contributed to a decrease in lower-body power of up to ~31%, an increase of the muscle damage biomarker creatine kinase (CK) of up to ~450%, as well as negative disturbances to mood, and testosterone and cortisol [4]. From a player fatigue perspective, these negative changes did not typically return to pre-match baseline levels until 48–72 hours post-match [4, 6].

Post-match fatigue is multidimensional [7], and can be quantified by objective (e.g. neuromuscular performance, cardio-autonomic, tissue biomarker tests etc.) and subjective (e.g. self-reported soreness, fatigue, mood etc.) methods and metrics [6, 8]. During the competitive season, the deleterious effects of fatigue decrease the player's readiness to train within the constraints of the typical week to week match-play microcycle. Pre- and post-match monitoring of each player's fatigue state is therefore important in the in-season period to identify if players are or are not returning to their individual baseline state between matches [9]. For support staff, this necessitates applying monitoring systems that quantify and integrate the various dimensions of fatigue, and the subsequent recovery process. This can then inform decision support systems to maintain or adjust training prescription (e.g., frequency, intensity, duration), and/or implement additional or specific recovery interventions [10–12].

Whilst the importance of quantifying training and competition loads and fatigue in rugby has been established [13], there have been limitations in prior attempts to define fatigue in the context of the unique demands of rugby (i.e., the frequency and intensity of collisions and locomotor actions) [14]. Conceptual models and definitions are important as they assist in developing understanding of the phenomena under investigation, the potential causal structure of relationships between variables, and to aid in the explanation of the phenomena by integrating background knowledge [3]. However, investigations of fatigue in rugby have been limited to primarily identifying changes in fatigue markers in isolation [4] and determining the effects of specific strategies in minimising fatigue and/or promoting recovery [12, 14, 15]. When attempts have been made to operationally define fatigue in rugby these definitions have been taken from prolonged endurance cycling exercise [16], which is a different exercise modality with divergent external and internal loads to that of rugby. Therefore, there is currently no agreed upon specific conceptual definition of fatigue in rugby. Given the unique contribution of physical collisions and variable locomotor efforts, a conceptual definition is important for the development and refinement of monitoring systems.

There are multiple methods and associated metrics used to quantify various components of fatigue that have been explored in the literature [8, 14], although there may be differences between the research literature and those employed in practice (i.e. the research to practice 'gap') [17]. West et al. [18] surveyed 12 support staff from each of the English Premiership professional rugby union clubs, who rated the perceived importance of multiple monitoring tools

of external load, internal load, and fatigue. Here, the monitoring of fatigue through objective (e.g. neuromuscular function), and subjective (e.g. self-reported ratings) methods was of comparable importance to GPS-derived external load [18]. However, given the focus on one code and level of competition, it is unclear whether these measures are rated similarly across rugby and differing levels of competition [18]. Moreover, several barriers for implementation, including the validity and reliability of the equipment, a lack of equipment and resources, and consensus on best practice use were identified. Given these barriers are perceived to exist in this professional setting, it is expected such barriers, and others, would limit the use of these methods in resource constrained situations, such as in the semi-professional or junior/developmental setting. It is not known which methods and metrics support staff are choosing to use to quantify fatigue across rugby and levels of professionalism.

The Delphi method gathers opinions and knowledge of Subject Matter Experts (SME) through multiple rounds of either questionnaires, surveys, or workshops to reach expert group consensus. Broadly, the Delphi method allows researchers to derive insights from SME on contentious issues, and by leveraging these insights, reduce the often-observed research to practice gap [19–22]. Within sport, the Delphi method has been used previously to achieve consensus on golf tournament preparation [20], return to play criteria for hamstring injuries [22], a youth sport specialisation definition [23], and sport-related concussion guidelines [24]. Whilst participation numbers in Delphi studies vary, within these Delphi studies in sport the total sample sizes of SME range from 17 to 58 participants [20–24].

Despite the increasing use of fatigue monitoring methods, and the known usefulness of quantifying player fatigue post-match, the use and importance and feasibility to implement of the various methods and metrics is not presently known. Further, there is, to our knowledge, currently no broadly agreed definition of fatigue in rugby, and no resource which is currently available to select fatigue monitoring methods or metrics in rugby. Therefore, the aims of this investigation were to develop a definition of fatigue in rugby, to establish agreement with this common definition, and to develop a resource of a fatigue monitoring system containing methods and metrics support staff use and consider important and feasible to implement to quantify fatigue. The final fatigue monitoring system included only those items which rated highly for importance to implement and feasibility to implement. To achieve these aims, the Delphi method was used to obtain consensus from a group of volunteer SME on fatigue monitoring methods and metrics drawn from neuromuscular performance tests, cardio-autonomic tests, tissue biomarkers, and self-reported measures. Whilst fatigue can occur in response to training, this study focused on fatigue as the external loads of match-play (collision and locomotor loads) represent the largest fatiguing stimulus [4, 13].

## Methods

### Experimental approach to the problem

The present study used a modified Delphi method to establish group-based consensus [25]. Online questionnaires were used in an attempt to obtain the opinions of SME to achieve consensus on a rugby-specific definition of fatigue, and the methods and metrics for quantifying fatigue in rugby. Each round of the questionnaire was open for four weeks in 2021, with rounds separated by a four-week period.

### Ethical approval

SME were provided with an information document and informed consent was then provided by all SME prior to commencing the questionnaire. The SME provided consent digitally by agreeing that they had read the documentation and were prepared to undertake the study.

Potential participants could not begin the questionnaire without providing their informed consent prior. Ethical approval for the study was obtained by the University of the Sunshine Coast Human Ethics Committee (HRECS211532).

## Subjects

The participants were SME from rugby who were invited to participate using a social media advertisement, through professional networks of the investigators, and directly through publicly available Email addresses and contact information. This is consistent with participant recruitment strategies in prior Delphi studies (e.g., Mokkink et al. [26]).

To ensure that a diverse range of stakeholder groups were represented, SME from both academic research institutions and rugby support staff (e.g., sports scientists, strength and conditioning coaches, performance analysts, technical/tactical coaches) who were currently working in rugby were eligible. Inclusion criteria were either actively researching and/or holding a current position within rugby as a research academic and/or support staff. For research academics, a background in scientific publishing in monitoring of match-demands and/or fatigue in rugby ($\geq$3 publications) was required, which is consistent with prior Delphi studies in sport [20, 21]. The SME could indicate that they work with more than one level of professionalism, or work with more than one sport (provided they held at least one role within rugby), or indicate they held multiple roles.

## Questionnaire development

The questionnaire developed by the investigators for round one included common fatigue monitoring methods and metrics to be rated (termed 'items') which were based on the extensive practical and research experience of the investigators alongside findings from previous reviews and applied studies in rugby for the neuromuscular performance, cardio-autonomic, tissue biomarker, and self-reported domains [4, 6, 8, 15, 27–31]. For example, within the neuromuscular performance domain, an included item was "countermovement jump (force [N]/power [W])", indicating force and power metrics within the countermovement jump test measured directly through force plates/platforms. The initial questionnaire was piloted through three rounds of revisions and agreed to by the investigators. Once finalised, round one of the questionnaire included 36 questions which covered broad categories: 1) demographic information, 2) open-ended questions related to how SME consider fatigue processes, and 3) rating the methods and metrics they consider important and feasible to implement to quantify fatigue. Round two of the questionnaire was developed based on SME responses and suggestions in round one.

## Procedure

A web-based platform was used to administer the questionnaire in each round (Qualtrics, Utah, USA). The questionnaire is available at https://osf.io/p7vud/. The SME were asked to score the relative importance of each post-match fatigue item within their sport and level of competition using a 5-point Likert scale (1: "Not at all important", 2: "Slightly important", 3: "Moderately important", 4: "Very important", and 5: "Extremely important"). The SME were also asked to score the feasibility to implement of each fatigue item to players within their sport and level of competition using a 5-point Likert scale (1: "Extremely difficult", 2: "Somewhat difficult", 3: "Neither easy nor difficult", 4: "Somewhat easy", and 5: "Extremely easy"). The number of questionnaire rounds was not set *a priori*.

**Questionnaire round one.** Round one of the Delphi questionnaire was available for completion for four weeks, and once opened, SME received a personalised reminder of the closing data seven days prior to the questionnaire round ending.

The cut-off values for consensus within the Delphi literature range from 55% to 100% agreement, and studies using similar designs to the present study in sport research have used a consensus criterion of ≥67% agreement across two score categories on a five-point Likert scale [20, 21, 26]. Therefore, consensus was reached if there was ≥67% agreement in two adjacent scale categories (e.g. 1 and 2 ["not at all important" and "slightly important"], 4 and 5 ["very important" and "extremely important"], etc.) [20, 26]. Once consensus was achieved for a given response item, the median score (± interquartile range [IQR]) for that item were determined. Items which achieved consensus in round one were not included to rate in round two. If ≤67% agreement was reached for a given response item for importance or feasibility to implement in round one, it was included in round two of the questionnaire to rate for that feature (i.e., importance or feasibility to implement). The consensus percentage and the median rating for those items were also presented to SME in the round two [26]. The SME in round one were asked to provide feedback on items that they thought needed to be changed or included in the questionnaire.

In addition to the rating of items, SME in the first round answered a series of open-ended response questions:

1. "*On what days relative to the match-day (MD), do you (or would you) consider it important to monitor the athlete's fatigue state?*"

2. "*In the context of your sport (i.e., rugby code), how do you (or would you) currently define athlete fatigue?*"

3. "*In the context of your sport (i.e., rugby code), which specific 'types' of fatigue (e.g. neuromuscular, metabolic, perceptual, biochemical, autonomic etc.) do you consider it important to monitor (please specify more than one 'type' if it is applicable)?*"

The responses to the open-ended questions 2 and 3 in round one were analysed to identify common themes among the responses which were then merged into one definition. This draft definition was discussed and amended for clarity (where necessary), before being agreed to by the investigators before round two. This definition was then rated for agreement consensus across a three-point Likert scale (agree/unsure/disagree) by the SME in round two.

**Questionnaire round two.** Prior to round two, the SME were sent a brief synopsis explaining the results of round one. Once round two had opened, the SME received a personalised reminder of the closing date seven days prior to the questionnaire round ending. Within the round two questionnaire, the median rating for items which did not achieve consensus for importance or feasibility to implement was provided in the body of the text explaining each response item. If an item reached consensus for importance or feasibility to implement but not for the other feature (e.g., for importance but not for feasibility to implement, and *vice versa*) in round one, it was only included in round two for the feature which did not achieve consensus.

**Fatigue monitoring system.** The items which obtained consensus and were rated highly (≥3/5) for both importance and feasibility to implement were included in the final fatigue monitoring system. Included in this fatigue monitoring system were considerations for data collection and analyses drawn from relevant literature in the neuromuscular performance, cardio-autonomic, and subjective (i.e., self-report assessment) domains.

## Data analysis

Following the data collection periods in round one and round two, the SME responses were exported from the Qualtrics platform to Microsoft Excel (Microsoft Corporation, Washington,

USA) and the responses were aggregated for statistical analysis. Data were analysed to determine the median (± IQR) for each item, and the consensus agreement percentage across each two adjacent scale categories. The open-ended responses to questions 2 and 3 (above) were analysed in NVIVO (Version 12, QSR International Pty Ltd) to calculate word clouds based on the 30 most used words and phrase responses.

## Results

### Participants

Fig 1 is a flowchart of the study questionnaire rounds. A total of 64 SME responded to round one, however, after removal of SME who answered less than 50% of the questions, provided dummy responses (e.g., incoherent or single word responses), or who answered "no" to providing consent to collecting their data (n = 22), 42 responses to round one were collected (66% of total respondents). For round two, 23 of the 42 SME from round one (55%) completed the questionnaire.

The SME in round one (n = 42) were aged 35.0 (± 7.4) years with 11.7 (± 8.3) years of experience within their sport. The SME indicated they worked in different levels of professionalism (Fig 2A), and worked in different codes of rugby (Fig 2B).

The SME were employed in the United Kingdom (43%), Australia (36%), Ireland (7%), New Zealand (5%), France (2%), Russia (2%), South Africa (2%), and an unspecified location (2%). There was a skew with SME being from English-speaking countries due to the questionnaire being available only in English. The SME indicated they were working as a Strength and Conditioning Coach (28%), Research Academic (26%), Sports Scientist (24%), High

**SME Recruitment**
- Social media advertisement
- Professional networks of the investigators
- Publicly available Email addresses and contact information

**Round One**
- Questionnaire open for four weeks
- Likert-based ratings and open-ended responses
- 64 respondents
- n = 42 (following removal of non-responders [refer to text for further details])

**Round Two**
- Questionnaire open for four weeks
- Likert-based ratings
- n = 23 (of 42 from Round One; no non-responders within this group)

**Fig 1. The flowchart of the questionnaire rounds including recruitment, the content of the questions, and the number of respondents to each round of the questionnaire.** SME—Subject matter experts.

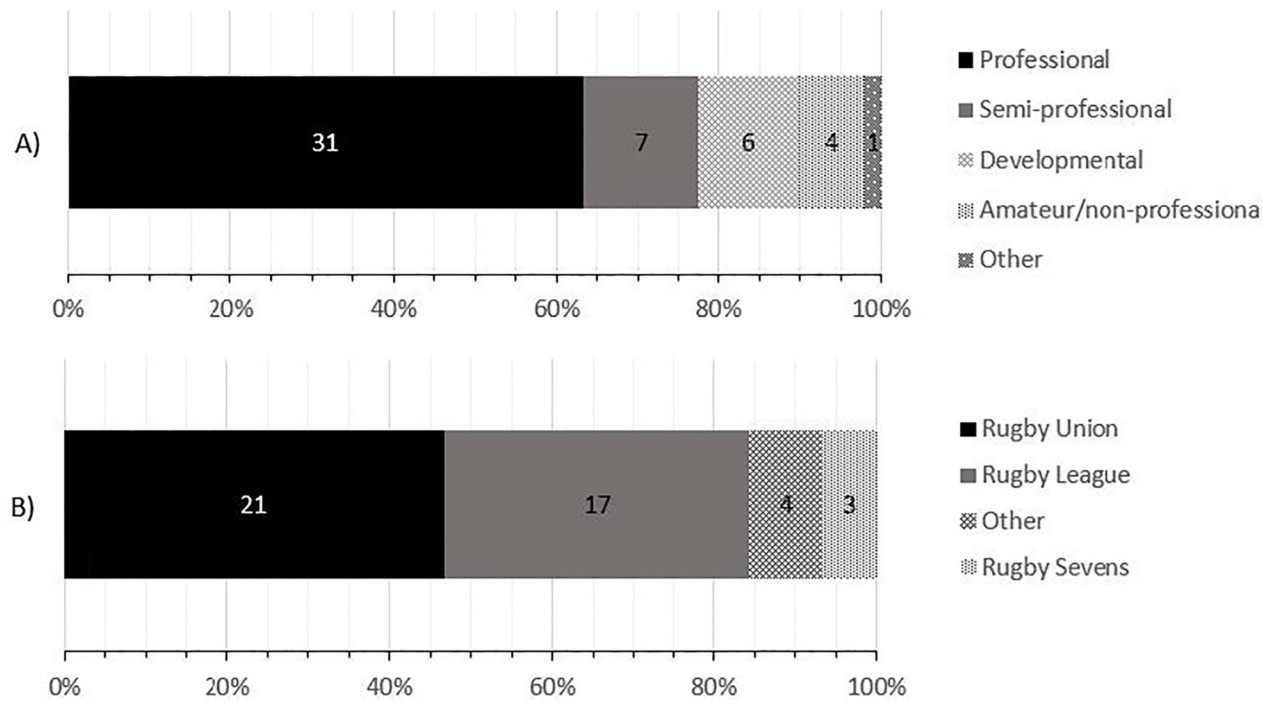

**Fig 2. The count of SME (n = 42) responses in round one by A) level of professionalism and B) the rugby football code they indicated they work in.** For A) other refers to holding a position at Olympic-level sport (considered separate to amateur sport), and B) other refers to also working in an Olympic sport (e.g., track and field). SME were free to choose more than one level of professionalism (e.g., professional and semi-professional) and sport (e.g. rugby league and rugby union) if they worked across multiple roles or sports. Values within the bars are counts of SME responses, and cumulative count percentage is presented on the x axis.

Performance Manager (12%), Technical/Tactical Coach (6%), and as a Performance Analyst or Performance Director (2%).

## Analysis

**Round one.** In responding to the open-ended question; "*On what days relative to the match-day (MD), do you (or would you) consider it important to monitor to monitor the player's fatigue state*?", SME commonly monitored fatigue on MD+2 followed by MD-2 which is are 48 hours pre-, and 48 hours post-match, respectively (Fig 3). Conversely, monitoring of fatigue on the MD and MD+1 had the lowest frequency (Fig 3).

The SME indicated that they believed there were different 'types' of fatigue present in rugby, and further defined fatigue in rugby as an effect on a broad range of responses for various domains and subdomains. The 30 most used words and phrase responses to questions 2 and 3 are presented as 30-word word clouds (S1A and S1B Fig, respectively).

In total, 22 items were rated in round one for importance and for feasibility to implement. However, due to the low number of SME (n = 3) who rated blood biomarkers for importance and feasibility to implement, these items were not included in the analysis or in round two of the questionnaire. This reduced the response items to 15 for importance, and 15 for feasibility to implement (for a total of 30 ratings) in round one. Of these, 13/15 (87%) achieved ≥67% consensus for importance (Fig 4A), and 10/15 (67%) achieved ≥67% consensus for feasibility to implement (Fig 4B). These items are presented and ranked by the median (± IQR) rating for importance (Fig 4A), and feasibility to implement (Fig 4B) for the three domains.

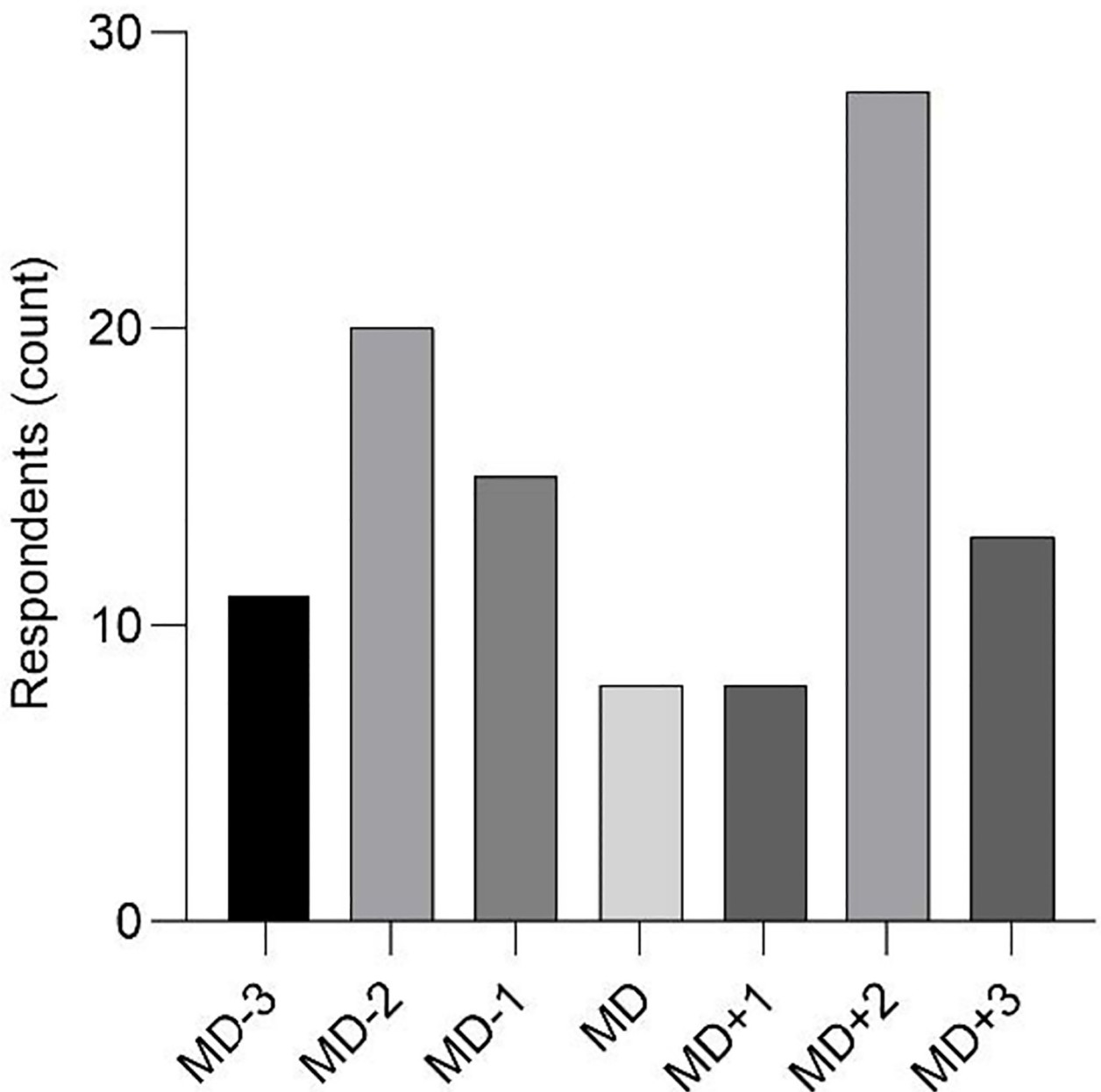

**Fig 3. Count of SME who responded to** *"On what days relative to the match-day (MD), do you (or would you) consider it important to monitor to monitor the athlete's fatigue state?"*. The SME were free to select more than one selection if they considered it important.

Based on the suggestions from SME in round one and agreement by the investigators, seven new items were introduced for round two. For example, a SME suggested "sleep quantity" was a self-reported measure that they consider to be important in quantifying fatigue post-match, and it was included to be rated in round two.

**Round two.** Common responses to the open-ended questions 2 and 3 (S1A and S1B Fig) in round one were analysed to derive a common definition of fatigue:

"*Fatigue in rugby refers to a reduction in performance-related task ability which is underpinned by time-dependent negative homeostatic changes within and between cognitive (e.g.,*

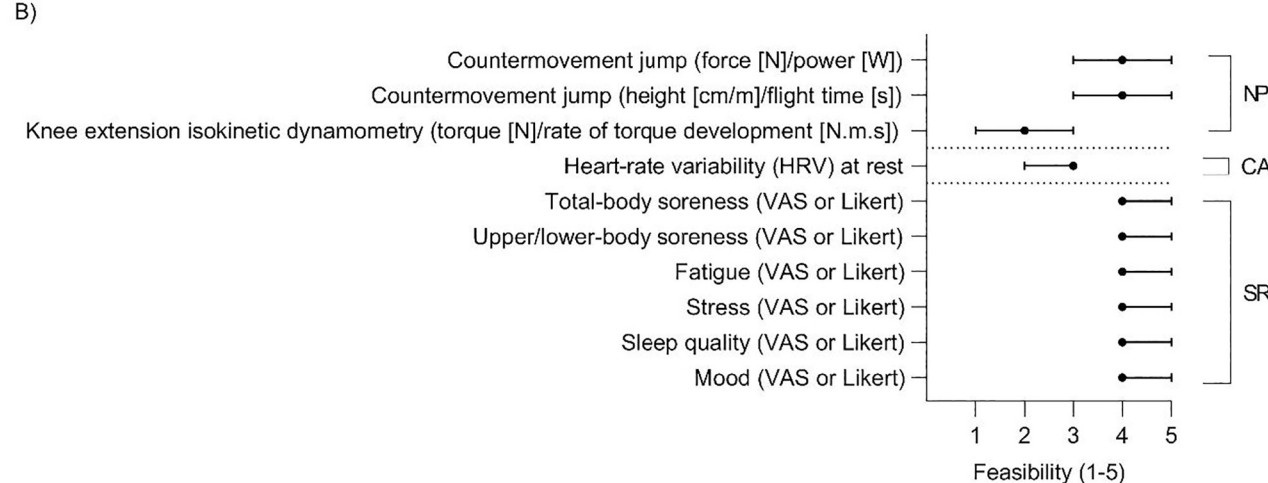

**Fig 4.** The SME ratings (median ± interquartile range [IQR]) of A) importance, and B) feasibility to implement of the items which achieved consensus in round one of the Delphi questionnaire for the neuromuscular performance (NP), cardio-autonomic (CA), and self-reported (SR) domains. For importance, the scale ranged from 1 –not at all important, to 5—extremely important. For feasibility to implement the scale ranged from 1 –extremely difficult to 5 –extremely easy. If end of the IQR error bar(s) is not visible it is due to this being located at the median. N—Newtons, VAS—Visual Analogue Scale, W—Watts.

*tiredness, lethargy), neuromuscular (e.g. force, power), perceptual (e.g. perceived exertion), physiological (e.g. cardio-autonomic, muscle damage), emotional (e.g. motivation), and technical/tactical (e.g. positioning, skill execution) domains.*"

The SME rated their level of agreement with this statement and 96% (22/23) of round two SME agreed with this definition of fatigue in rugby, with one SME rating their agreement as 'unsure'.

Of the 21 total item ratings for importance and feasibility to implement which were presented in round two, 10 achieved consensuses, with those items ranked by median (± IQR) for importance (Fig 5A), and feasibility to implement (Fig 5B) over the three domains.

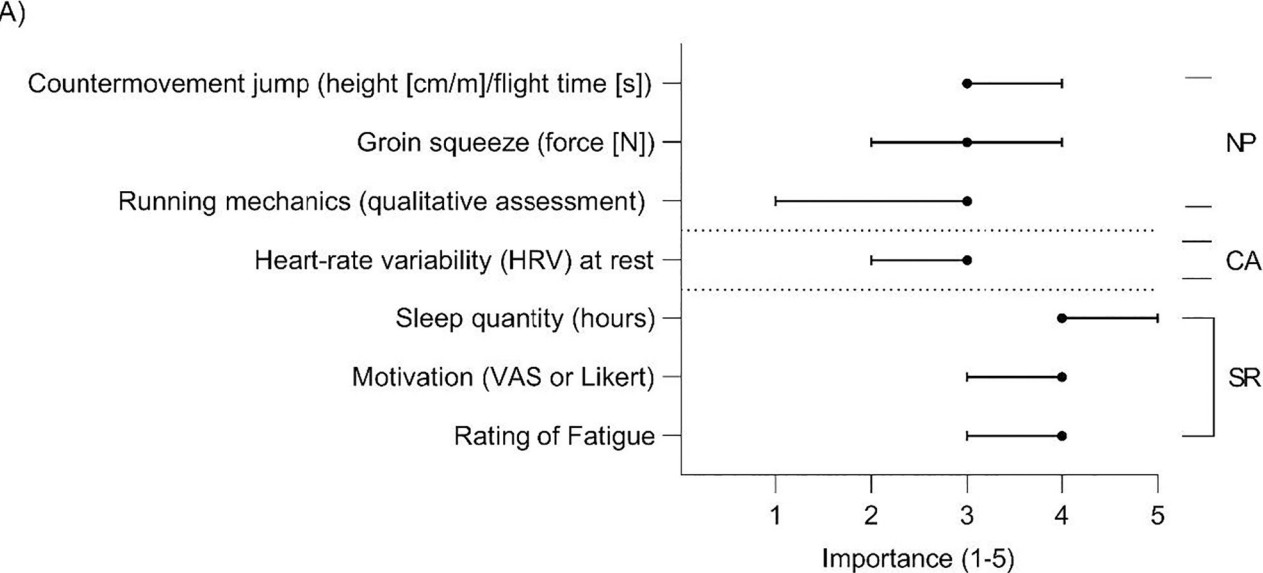

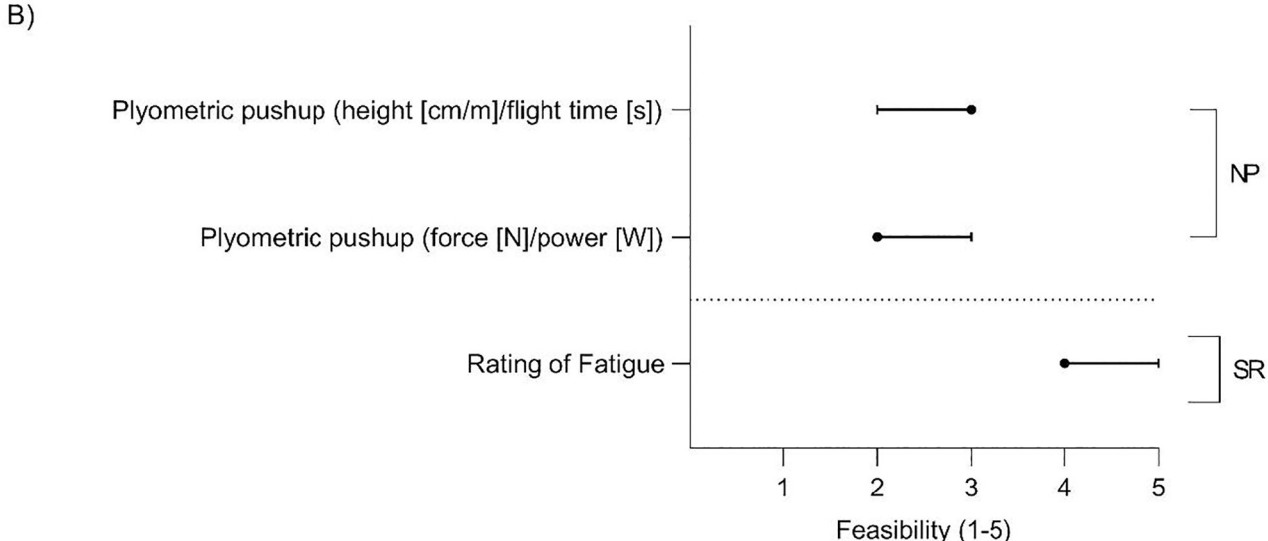

**Fig 5. The SME ratings (median and interquartile range [IQR]) of A) importance, and B) feasibility to implement of the items which achieved consensus in round two of the Delphi questionnaire for the neuromuscular performance (NP), cardio-autonomic (CA), and self-reported (SR) domain measures.** For importance, the scale ranged from 1 –not at all important, to 5—extremely important. For feasibility to implement the scale ranged from 1 –extremely difficult to 5 –extremely easy. If end of the IQR error bar(s) is not visible it is due to this being located at the median. N—Newtons, VAS—Visual Analogue Scale, W—Watts.

The items which did not achieve consensus through the two rounds of the Delphi questionnaire is presented (S1 Table). A third round of the Delphi questionnaire was not undertaken due to the likely drop off in SME participation (as occurred between round one and round two), and the 96% agreement with the fatigue definition presented in round two.

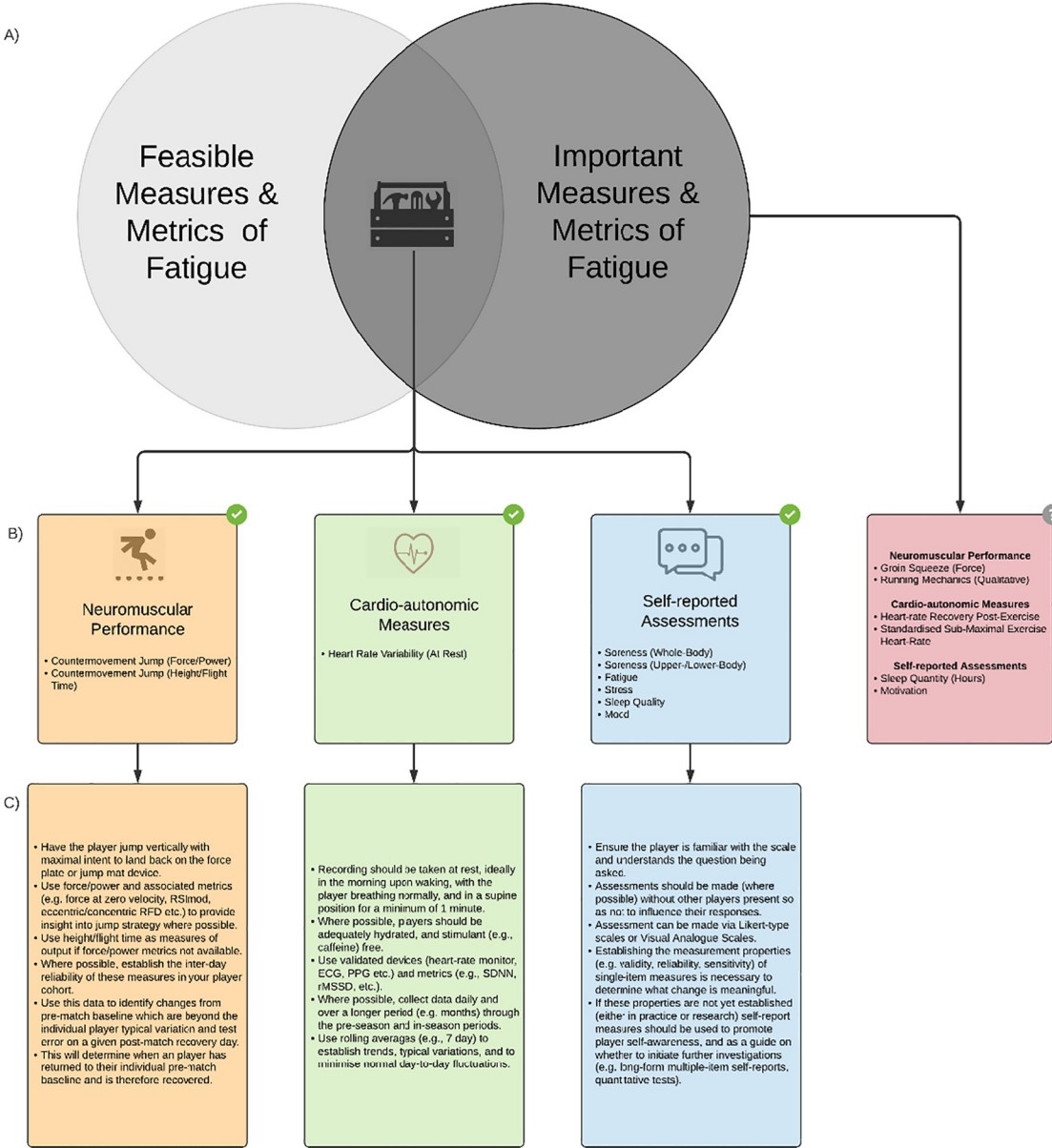

**Fig 6.** A graphical representation of a post-match player fatigue monitoring system based on SME ratings for A) important and feasible measures and metrics. The fatigue monitoring system includes B) a 'toolbox' of items that achieved consensus and rated highly (≥3/5) for importance and feasibility to implement in the rugby football codes for objective (i.e. neuromuscular performance [32–34], cardio-autonomic [35, 36]), and subjective (i.e. self-report assessment [5, 37, 38]) domains. Items which achieved consensus and rated highly for importance but not for feasibility to implement are represented by the arrow linking to the box to the right of the Venn diagram. Included within the graphical representation are C) considerations for testing and analysing data for the domains of these measures and metrics (see [5, 32, 33, 35–40] for further assessment considerations). ECG—echocardiogram, PPG—photoplethysmography, rMSSD—square root of the mean sum of the squared differences in R-R intervals, RFD—rate of force development, RSImod—reactive strength index modified, and SDNN—standard deviation of all normal-to-normal intervals.

A graphical representation of a fatigue monitoring system which includes the items which achieved consensus and were rated highly for importance and feasibility to implement (Fig 6A and 6B). The graphical representation includes considerations for testing and data analyses (Fig 6B and 6C).

## Discussion

The aims of this study were to develop a definition of fatigue in rugby and to obtain SME agreement on this common definition of fatigue in rugby, and to develop a resource of a fatigue monitoring system containing the fatigue monitoring methods and metrics that are considered important and feasible to implement. The SME agreed upon a definition of fatigue in rugby developed from the round one responses which achieved almost perfect agreement in round two. Moreover, of the 51 items the SME rated for importance and feasibility to implement in rounds one and two, 33 achieved consensus for neuromuscular performance, cardio-autonomic, and self-report measures. A graphical representation of a fatigue monitoring system was developed based on the items which obtained consensus and were rated as of high importance and feasibility to implement. This representation includes data collection and analyses considerations.

A fundamental contribution of the present study was a consensus agreement on a definition of fatigue in rugby which was developed based on the knowledge and perspectives of the SME. This definition demonstrates that researchers and practitioners with extensive expertise in rugby consider fatigue to be a time-dependent acute and negative change which is multidimensional. Consequently, fatigue in rugby is purported to include and effect cognitive, neuromuscular, perceptual, physiological, emotional, and technical/tactical domains [7, 14, 30, 41]. Prior research to define fatigue in rugby has been limited to the determination of the effects of muscle damage from both exercise and collisions [15, 42], and the measurement of various objective and subjective variables as markers of fatigue in isolation [4, 14, 28–30]. Previous attempts to operationally define fatigue in rugby have been limited to the ". . . sensations of tiredness and associated decrements in muscular performance and function." [13, 14]. However, this definition has been adopted from research in endurance cycling [16], and appears to have limited scope in rugby given the SME in the present study considered fatigue as a negative effect ". . . within and between cognitive, neuromuscular, perceptual, physiological, emotional, and technical/tactical domains". Therefore, we believe the present study is the first to seek and achieve agreement from SME on a common definition of fatigue whilst considering the distinctive characteristics of rugby.

It is fundamental to this study that this definition was developed using the knowledge and perspectives of the SME who then agreed to its content. Further, the approach of the present study is not unusual, as incorporating SME feedback through the Delphi process to develop a definition and then seek agreement with that definition has been implemented in other areas of sport including running-related injuries [43], and early sport specialization [23]. Fatigue in exercise and sport is complex and has been researched for a variety of symptoms and context-specific operational definitions in other literature (e.g., central/peripheral, mental/physical, performance/perceptual etc.) [7, 44, 45]. In the present study the SME agreed upon a definition which includes performance, perceptual, physiological, emotional, and cognitive aspects [7, 41, 46]. Further research can refine this agreed upon definition, as necessary, and determine the relative contributions of the presented components of fatigue (and their interactions) in rugby.

When considering neuromuscular performance measures, our results agree, in part, with those of West et al. [18] who found that neuromuscular performance testing was considered important in rugby, and Taylor et al. [47] who identified that jump tests are common in rugby. The measurement of force and power in the countermovement jump, typically measured directly using a force plate/platform [28], was identified as an important and feasible to implement measure of neuromuscular performance fatigue in rugby. Jump height or flight time in comparison were rated of lower importance but of similar feasibility to implement to force or

power measures. Conversely, in the present study, monitoring fatigue through the plyometric pushup, knee extension isokinetic dynamometry, and jump height measured by a validated smart-phone application [48] were considered of low importance and/or difficult to implement. It is expected that this indicates a number of perceived barriers such as the inadequate reliability and sensitivity of plyometric pushup force and power variables [49], the space, setup and cost requirements of isokinetic dynamometers [50], and the time to analyse the data from each of these tests [18]. Which other common and specific barriers limit the application of fatigue monitoring in rugby are yet to be determined.

In the present study, SME consistently rated self-report measures highly when considering their importance and feasibility to implement. For whole-body or differentiated (i.e. lower-, or upper-body) soreness, fatigue (Likert/VAS, or Rating of Fatigue [51]), sleep, mood, and motivation, items were consistently rated as being either moderately or very important, and being somewhat easy to implement or higher. The high ratings are potentially to be expected, given that these measures are easily recorded and require minimal equipment and human resources [37]. Other research has critiqued the use of single-item (i.e. question) self-reported measures in sport [38]. Jeffries et al. [38] systematically reviewed multiple-item and single-item self-reported questionnaires, and observed that single-item self-report measures, such as those included in the present study, have been used without undergoing formal validation. Therefore, as the measurement properties (e.g. test-retest reliability, sensitivity [signal to noise ratio] etc.) of these measures have not been determined, the subsequent conclusions are difficult to interpret [38]. As the results of the present study indicate considerable weighting is currently attributed to self-reported/subjective measures as indicators of fatigue, further validation of these items in the fatigue monitoring context is required.

There are several potential limitations in the present study. Firstly, the opinions of the SME in this study may not be representative of other research academics and support staff who were not recruited, and who could provide different ratings. Conversely, in the present study a range of SME volunteered from all the rugby codes to provide a sample size which exceeds that of comparable studies in rugby union (n = 12) [18], and golf (n = 36) [20]. This should have ensured that the aggregated perspective determined from the range of SME who were drawn from a worldwide cohort was representative and generalisable in rugby. Whilst there was a clear bias in that SME were recruited from English-speaking countries, this likely reflects the questionnaire being available in English and the skewed distribution in popularity of rugby in these countries.

Secondly, there was large heterogeneity in the experience of the SME, and their responses are only representative at the time of responding. These responses may therefore change over time with experience and the constant changes in technology for player fatigue monitoring. These factors mean that items which did not achieve consensus (S1 Table), or were not rated (e.g., tissue biomarkers) may achieve consensus in a separate cohort or study. Further, although the definition achieved near perfect agreement (96%), the SME did not have the option to provide suggested changes to the fatigue definition in round two.

Thirdly, the use of >3/5 for both importance and feasibility as the threshold for inclusion in the final monitoring system was chosen somewhat arbitrarily. However, should a higher (or lower) threshold be of interest in a specific situation, the findings of the present study allow support staff to determine that threshold for importance and feasibility on an individual basis. Finally, whilst the monitoring system has been developed from the responses of the SME, the final list of items has not been empirically tested (either qualitatively or quantitatively) against other potentially relevant items or systems.

In conclusion, using two rounds of a questionnaire with a modified Delphi study, consensus was obtained on a definition of fatigue contextualised to the multidimensional components

of rugby. This defines fatigue in rugby as a reduction in performance-related task ability which is underpinned by time-dependent negative homeostatic changes within and between cognitive, neuromuscular, perceptual, physiological, emotional, and technical/tactical domains. Further, consensus was obtained on the items that are considered important and feasible to implement by the SME to quantify fatigue. These methods and metrics quantify different components of fatigue, providing further agreement with the multidimensional components of fatigue, and the corresponding definition of fatigue that was agreed to in the present study. Finally, a fatigue monitoring system of objective (i.e., neuromuscular performance, cardio-autonomic) and subjective (i.e. self-report) methods and metrics was developed which should be used to guide the selection of methods and metrics for the monitoring of fatigue in rugby.

## Practical applications

From the results of the present study, countermovement jump force and power, heart-rate variability (at rest), self-reported soreness (total or upper/lower-body soreness), fatigue, stress, sleep quality, and mood rated highly for both importance and feasibility to implement and were included in the monitoring system (Fig 6B). For practitioners who do not have access to force plates/platforms, countermovement jump height and flight time could be considered as an alternative, with a lower level of perceived importance and sensitivity to detect fatigue [27]. The monitoring system presented in the present study (Fig 6) includes the multidimensional components for the neuromuscular, cardio-autonomic, and self-report domains [4, 8, 37]. Further, the findings (Fig 3) outline that fatigue monitoring should ideally occur prior to and following the match-day or tournament days (in the case of a multiple day tournament), to establish the individual players change from baseline. For other practitioners who consider feasibility to implement to be achievable and thus importance is considered in isolation, the groin squeeze test [52], running mechanics (assessed qualitatively), and/or heart rate during a standardised, submaximal run [53] could be considered as additions in a fatigue monitoring system (Fig 6B). Practitioners should tailor these findings to their individual circumstances, including potential barriers to implementing monitoring such access to equipment, staffing, financial resources, and scheduling.

For the measures which achieved consensus in this study, practitioners should always consider measurement criteria such as the validity, reliability, accuracy, and sensitivity when selecting which to implement [8, 21, 33, 34]. For example, quantifying the typical variation and error for a specific variable on a specific post-match recovery day (i.e. MD+1, MD+2 etc.) allows for identifying changes from baseline which extend beyond this typical variation for the variable and the error of the test [21, 33]. Further research is necessary to establish and compare the sensitivity of measures agreed upon in the present study for identifying a usable fatigue value, whilst considering external load differences between players and potential mediators of the load to fatigue relationship (e.g., strength, fitness [54]).

## Supporting information

**S1 Fig.** Responses to questions A) "*In the context of your sport (i.e. rugby code), how do you (or would you) currently define player fatigue*?", and B) "*In the context of your sport (i.e. rugby code), which specific 'types' of fatigue (e.g. neuromuscular, metabolic, perceptual, biochemical, autonomic etc.) do you consider it important to monitor (please specify more than one 'type' if it is applicable)*?" presented as 30 word clouds which are the 30 most commonly used words and phrases used in the responses in round one SMEs (n = 42).
(DOCX)

**S1 Table. Characteristics of items which did not achieve consensus (≥67% agreement) across two adjacent categories in the two round Delphi study including the largest consensus agreement obtained, the median rating, and the round in which agreement and rating was obtained.**
(DOCX)

## Acknowledgments

The investigators would like to thank the participants for providing their valuable responses for which this manuscript is based.

## Author Contributions

**Conceptualization:** Mitchell Naughton, Scott McLean.

**Investigation:** Mitchell Naughton, Tannath Scott, Dan Weaving.

**Methodology:** Mitchell Naughton, Tannath Scott, Dan Weaving, Colin Solomon, Scott McLean.

**Supervision:** Tannath Scott, Dan Weaving, Colin Solomon, Scott McLean.

**Visualization:** Mitchell Naughton.

**Writing – original draft:** Mitchell Naughton.

**Writing – review & editing:** Mitchell Naughton, Tannath Scott, Dan Weaving, Colin Solomon, Scott McLean.

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
