## [Decision Letter · Decision Letter 0]

24 Jan 2023

PONE-D-22-28342Defining fatigue and quantifying post-match fatigue in the rugby codesPLOS ONE

Dear Dr. Weaving,

Thank you for submitting your manuscript to PLOS ONE. After careful consideration, we feel that it has merit but does not fully meet PLOS ONE’s publication criteria as it currently stands. Therefore, we invite you to submit a revised version of the manuscript that addresses the points raised during the review process.

ACADEMIC EDITOR:Dear Authors, your manuscript has been revised by one expert in the fields, which retrieved some minor issues you should consider while revising the work. Please submit your revised manuscript by Mar 10 2023 11:59PM. If you will need more time than this to complete your revisions, please reply to this message or contact the journal office at plosone@plos.org. Please include the following items when submitting your revised manuscript:A rebuttal letter that responds to each point raised by the academic editor and reviewer(s). You should upload this letter as a separate file labeled 'Response to Reviewers'.A marked-up copy of your manuscript that highlights changes made to the original version. You should upload this as a separate file labeled 'Revised Manuscript with Track Changes'.An unmarked version of your revised paper without tracked changes. You should upload this as a separate file labeled 'Manuscript'.If applicable, we recommend that you deposit your laboratory protocols in protocols.io to enhance the reproducibility of your results. Protocols.io assigns your protocol its own identifier (DOI) so that it can be cited independently in the future. For instructions see: https://journals.plos.org/plosone/s/submission-guidelines#loc-laboratory-protocols. Additionally, PLOS ONE offers an option for publishing peer-reviewed Lab Protocol articles, which describe protocols hosted on protocols.io. Read more information on sharing protocols at https://plos.org/protocols?utm_medium=editorial-email&utm_source=authorletters&utm_campaign=protocols.

We look forward to receiving your revised manuscript.

Kind regards,

Emiliano Cè

Academic Editor

PLOS ONE

Journal Requirements:

Reviewers' comments:

Reviewer's Responses to Questions

**Comments to the Author**

1. Is the manuscript technically sound, and do the data support the conclusions?

Reviewer #1: Yes

2. Has the statistical analysis been performed appropriately and rigorously? 

Reviewer #1: Yes

3. Have the authors made all data underlying the findings in their manuscript fully available?

Reviewer #1: Yes

4. Is the manuscript presented in an intelligible fashion and written in standard English?

Reviewer #1: Yes

5. Review Comments to the Author

Reviewer #1: The study entitled “Defining fatigue and quantifying post-match fatigue in the rugby codes” aimed to set up a definition of fatigue which could fit for a range of Rugby modalities, as well as determine the methods and metrics for fatigue assessment which experts consider important in the rugby daily-based routine.

I would like to congratulate the authors for the study which is really interesting and highly applicable for coaches, sports scientists, performance analysts, practitioners, academic researchers etc. Follow bellow minor details and few suggestions.

The first paragraph of the introductions is confusing, most likely due the excessive information inside the parentheses. Please, revise this paragraph.

The second paragraph of introduction (from line 65 to 82) and the sixth paragraph of the discussion (from line 423 to 448) are too long, please, split each one in two paragraphs.

I would suggest to the author to add a figure of the experimental design timeline. Despite well explained, the many steps of the study would be better visualized by the readers with a figure of the time domain and relevant information (e.g. sample size of each round).

There are several spots in the text with typo errors (lines 271, 272, 283, 355), please double check throughout the manuscript.

I’m not sure if I missed that in the supplementary files, however, would be interesting provide the questionnaires in their final form.

What was considered “dummy responses”? (line 245)

Finally, two points to the authors think about.

First, despite the descriptive nature of the study, would be interesting provide some kind of sample size calculation and/or statistical power. That would strengthen the paper.

Second, I don’t agree with “post-match fatigue quantification”. According to the first open-ended question, MD-2 is the second most frequent condition of fatigue assessment. That would not be a “pre-match fatigue” depend on the day of prior match? I suggest (even in the manuscript title) “quantifying match fatigue” only.

6. PLOS authors have the option to publish the peer review history of their article (what does this mean?). If published, this will include your full peer review and any attached files.

Reviewer #1: **Yes: **Fabio Milioni

---

## [Author Response · Author response to Decision Letter 0]

30 Jan 2023

Reviewer #1: The study entitled “Defining fatigue and quantifying post-match fatigue in the rugby codes” aimed to set up a definition of fatigue which could fit for a range of Rugby modalities, as well as determine the methods and metrics for fatigue assessment which experts consider important in the rugby daily-based routine.

I would like to congratulate the authors for the study which is really interesting and highly applicable for coaches, sports scientists, performance analysts, practitioners, academic researchers etc. Follow bellow minor details and few suggestions.

The authors wish to thank the reviewer for reviewing the manuscript and their recommendations and suggestions. The responses to each of these is presented below with the authors responses below the Reviewers comment. This has been included as an attachment as well where the author responses are highlighted in bold.

1) The first paragraph of the introductions is confusing, most likely due the excessive information inside the parentheses. Please, revise this paragraph.

This paragraph has been revised from Line 62 to Line 64 – 65 of the updated text and the text has been amended to remove the parentheses which fixes the issue that the Reviewer indicates.

2) The second paragraph of introduction (from line 65 to 82) and the sixth paragraph of the discussion (from line 423 to 448) are too long, please, split each one in two paragraphs.

These paragraphs have been revised and each split in to two paragraphs where it was most appropriate to do so, based on the content of these paragraphs.

3) I would suggest to the author to add a figure of the experimental design timeline. Despite well explained, the many steps of the study would be better visualized by the readers with a figure of the time domain and relevant information (e.g. sample size of each round).

Figure 1 is now a flow chart which details the timeline of the study, including recruitment information, content of the questionnaire in each round, and the number of participants.

4) There are several spots in the text with typo errors (lines 271, 272, 283, 355), please double check throughout the manuscript.

The authors have checked the manuscript for typological and grammatical errors and revised these (see manuscript). The authors have used the PLOS One guidelines (https://journals.plos.org/plosone/s/criteria-for-publication#loc-5) on using Standard English and therefore words such as “analyse” and “specialise” are used instead of “analyze” or “specialize” as would be the case for American English.

5) I’m not sure if I missed that in the supplementary files, however, would be interesting provide the questionnaires in their final form.

The authors agree however, due to the length of the questionnaire document, a supplementary file would not be appropriate. The survey content has been uploaded to a public open access repository, the Open Science framework (https://osf.io/p7vud/). This information has been included in the text (Line 184 of the updated text). 

6) What was considered “dummy responses”? (line 245)

A response was considered a dummy response if the respondent, for example, responded with an incoherent or one letter or word responses before moving on to the next question. These examples have been included in the text as examples.

7) Finally, two points to the authors think about.

First, despite the descriptive nature of the study, would be interesting provide some kind of sample size calculation and/or statistical power. That would strengthen the paper.

Sample sizes calculations to determine statistical power are generally not considered as part of Delphi studies (Trevelyan and Robinson, 2015) and samples can vary widely (e.g., 6 – 50 participants [Birko et al., 2015]). Importantly, as there are no statistical comparisons between the outcome measures (e.g., null-hypothesis significance testing, p values, effect sizes), determining statistical power to detect such differences is not relevant for this study. Whilst an increase of the size of the sample will increase generalisability of the resulting conclusions, the Delphi method requires a qualitative approach to the sample which relies on recruiting the subject matter experts in a given area. The authors believe we have achieved this by the targeted recruitment (i.e., directed emails, use of professional networks) and the inclusion/exclusion criteria. This ensured that the sample is homogenous with respect to their expertise, a recommendation of Trevelyan and Robinson (2015), and that the study has measured the consensus among these subject matter experts with a sample size which is comparable to other Delphi studies in sport.

8) Second, I don’t agree with “post-match fatigue quantification”. According to the first open-ended question, MD-2 is the second most frequent condition of fatigue assessment. That would not be a “pre-match fatigue” depend on the day of prior match? I suggest (even in the manuscript title) “quantifying match fatigue” only.

The authors agree with this suggestion, and this has been revised throughout the manuscript with the removal of “post-match” in the context of fatigue.

References

Birko, S., Dove, E. S., & Özdemir, V. (2015). Evaluation of Nine Consensus Indices in Delphi Foresight Research and Their Dependency on Delphi Survey Characteristics: A Simulation Study and Debate on Delphi Design and Interpretation. PloS one, 10(8), e0135162.

Trevelyan, E. G., & Robinson, N. (2015). Delphi methodology in health research: how to do it? European Journal of Integrative Medicine, 7(4), 423-428.

As mentioned by the Academic Editor, several additional requirements have been met in the updated manuscript. Firstly, the manuscript has been updated to adhere to the PLOS ONE style requirements, as mentioned by the Academic Editor. Secondly, information about the consent process has been included in the Methods section of the manuscript, and in the online submission information. Thirdly, captions have been included for the Supporting Information at the end of the manuscript. Finally, the reference list has been reviewed and amended to comply with the PLOS ONE style format.

---

## [Decision Letter · Decision Letter 1]

14 Feb 2023

Defining and quantifying fatigue in the rugby codes

PONE-D-22-28342R1

Dear Dr. Weaving,

We’re pleased to inform you that your manuscript has been judged scientifically suitable for publication and will be formally accepted for publication once it meets all outstanding technical requirements.

Kind regards,

Emiliano Cè

Academic Editor

PLOS ONE

Additional Editor Comments (optional):

Reviewers' comments:

Reviewer's Responses to Questions

**Comments to the Author**

1. If the authors have adequately addressed your comments raised in a previous round of review and you feel that this manuscript is now acceptable for publication, you may indicate that here to bypass the “Comments to the Author” section, enter your conflict of interest statement in the “Confidential to Editor” section, and submit your "Accept" recommendation.

Reviewer #1: All comments have been addressed

2. Is the manuscript technically sound, and do the data support the conclusions?

Reviewer #1: Yes

3. Has the statistical analysis been performed appropriately and rigorously? 

Reviewer #1: Yes

4. Have the authors made all data underlying the findings in their manuscript fully available?

Reviewer #1: Yes

5. Is the manuscript presented in an intelligible fashion and written in standard English?

Reviewer #1: Yes

6. Review Comments to the Author

Reviewer #1: I would like to congratulate the authors for the great work. I believe that will be very helpful and highly applicable in the sport science.

7. PLOS authors have the option to publish the peer review history of their article (what does this mean?). If published, this will include your full peer review and any attached files.

Reviewer #1: **Yes: **Fabio Milioni

---

## [Editor Report · Acceptance letter]

1 Mar 2023

PONE-D-22-28342R1 

Defining and quantifying fatigue in the rugby codes 

Dear Dr. Weaving:

I'm pleased to inform you that your manuscript has been deemed suitable for publication in PLOS ONE. Congratulations! Your manuscript is now with our production department. 

Kind regards, 

on behalf of

Professor Emiliano Cè 

Academic Editor

PLOS ONE